# The Status of Didactic Models for Heritage Education: A Systematic Review

Yenifer Karina Valencia Arnica [ID], Jackeline Lorena Ccasani Rodriguez [ID], Fabian Hugo Rucano Paucar *[ID] and Fabiola Talavera-Mendoza [ID]

Faculty Sciences of Education, Universidad Nacional de San Agustin de Arequipa, Arequipa 04000, Peru; yvalenciaa@unsa.edu.pe (Y.K.V.A.); jccasanir@unsa.edu.pe (J.L.C.R.); ftalaveram@unsa.edu.pe (F.T.-M.)

* Correspondence: frucano@unsa.edu.pe

**Abstract:** Heritage education is very important because it implies a holistic and transdisciplinary approach, where teachers must use resources and educational proposals that promote the conservation, appreciation, and care of heritage. The objective of this study is to analyze heritage education from a global perspective to identify didactic models, areas of educational action, categories, and typologies used in teaching and learning processes. A systematic review of the literature is proposed using the PRISMA methodology in three multidisciplinary databases by carrying out an exhaustive search with inclusion and exclusion criteria. The results highlight that teachers develop learning experiences focused on didactic models with pedagogical intervention in the classroom with formal action, focusing mainly on intangible heritage related to festive acts and rituals; so, they only achieve identity levels and do not reach the heritage levels necessary to create a legacy and promote the appreciation of cultural heritage. The use of digital educational strategies and resources is required to integrate real and simulated spaces with new educational and didactic approaches using virtual technologies. Consequently, this study implies that teachers need to develop digital skills to achieve more effective and meaningful heritage education.

**Keywords:** heritage education; secondary education; teaching; cultural heritage; systematic review

## 1. Introduction

Heritage education emerged as an educational necessity to preserve the value of cultural heritage [1]. Understanding heritage as a collection of tangible or intangible elements that are the legacy of a society's past, the aim is to preserve it in the present with the goal of continuing to pass it down to future generations [2,3].

The term heritage education originated in Brazil in the late 1960s and experienced exponential growth in 1972 thanks to UNESCO, which considers its task to be guaranteeing heritage protection through awareness-raising, communication, research, and education based on respect for and appreciation of cultural heritage [4–7].

For this reason, the Council of Europe recommends that the teaching of cultural heritage be carried out through interdisciplinary approaches and active methods and be reflected in school exchanges, curricula, student projects, and activities outside of school [8,9]. Similarly, in the Faro Convention of 2005, it is stated that every individual, whether on an individual or collective basis, has the legitimacy to enjoy their cultural heritage and contribute to its development [10]. Likewise, UNESCO considers education and heritage to be inseparable, as they allow communities to learn about their heritage assets, value and preserve them, and generate a sense of identity and citizenship with cultural values [11].

Currently, heritage education seeks to establish a connection between the subject and their heritage, establishing a relationship between their past and present and forming a responsible, critical, active, and reflective citizenship that is aware of the preservation and

transmission of its heritage [5,12]. Therefore, heritage education has become an important discipline that promotes the preservation and transmission of culture and heritage.

Heritage education has experienced notable qualitative and quantitative growth. However, an absence of systematic reviews of the literature on heritage education and its didactic and methodological implications in the area of social sciences has been observed. The author of [13], in his mixed-method study, has pointed out the lack of interest and deficient training of teachers in relation to heritage education. Another study has highlighted the need to incorporate subjects related to heritage education in initial teacher training [14], to apply active methodologies and educational strategies inside and outside the classroom, and to enhance heritage as a didactic resource that promotes active citizenship [15].

Regarding gaps, heritage proposals have been detected aimed not only at students but also at teachers to enrich their training and provide them with didactic tools or applications for teaching heritage [13]. Therefore, a holistic view of heritage should be explored through active teaching methodologies, such as role-playing [16]. In order to identify didactic models, forms of educational action, their impact on teaching, developed contents, and typologies in educational units or projects related to heritage education, a systematic literature review will be carried out. Working with heritage inside and outside the classroom contributes to the formation of critical and reflective citizenship, which represents an important contribution to the knowledge, valuation, and awareness-raising processes developed through teaching strategies used by teachers in formal, nonformal, and informal contexts [16].

## 2. Theoretical Framework

There are different approaches to heritage education, which include education with heritage, heritage education, education for heritage, and the education and heritage approach [17]. In the first approach, heritage is understood as a didactic resource, while in the second, the content is heritage and is combined with other curricular subjects in the collection of museums, sites, and heritage places. In the third approach, educational action is actively involved in heritage, and a guide is proposed on how to develop teaching and learning-related content. In the fourth approach, heritage is the content of learning, and people endow cultural value to heritage assets. In this sense, the analyzed studies focus on the education and heritage approach that corresponds to the fourth conceptualized approach.

Therefore, heritage education has a holistic and integrative vision [16], which is carried out through a multidirectional, multidisciplinary, systematic, and permanent educational process in formal, informal, and nonformal contexts [17], with the objective of disseminating, caring for, conserving, valuing, interpreting, activating, and transmitting heritage [18] for identification and heritage purposes. It is important to highlight that heritage education contributes to the sustainable, sociable, and institutional management of heritage [19], allowing the identification of didactic strategies that enable the development of knowledge and appreciation of the natural, social, and cultural environment, as well as the use and recognition of different forms of artistic representation and expression [16].

Heritage education is developed in three areas of educational action: the formal, informal, and nonformal levels [5,20]. The formal level focuses on the development of projects in which educational management takes charge of promoting heritage knowledge at different existing educational levels [12,21]. Regarding the informal level, it seeks to create situations and capacities that promote heritage education processes through visits to cultural centers such as museums, art galleries, workshops, and exhibitions [22]. At the nonformal level, innovations in teaching emerge, increasing the offer of heritage educational programs that help students understand their heritage [23,24].

Heritage education aims to create a symbolic appropriation between the subject and the heritage object through the process of heritagization, which consists of eight stages: knowing, understanding, respecting, valuing, sensitizing, caring, enjoying, and transmitting the heritage legacy [12,25]. This complex process is achieved through a network system

that involves the active participation of various actors, promoting values, interests, and meanings that ensure the preservation of heritage [26,27].

Heritage education develops identity links, whether individually or communally [5], and is linked to various aspects. Firstly, there is the training of teachers, highlighting the importance of teachers being innovative entities capable of developing proposals and practices in heritage education, considering heritage as curricular content to form critical and reflective citizenship, and promoting sustainable development from a symbolic and holistic vision [5].

Secondly, there are didactic proposals to create links between schools and heritage centers, developing civic–social competencies [28,29]. Action strategies are designed based on curricular contents to solve heritage conservation problems from their context [30,31], guiding students to exercise responsible attitudes of care for heritage assets, considering their degree, age, and level of education [32,33]. Similarly, there are didactic models that encompass a variety of adaptable and flexible teaching methods that teachers use to guide the educational process [34]. This takes into account various learning styles, student interests, the context in which they operate, and the actions of the teacher within the teaching process.

Finally, a third aspect arises that is related to heritage programs. Through the implementation of emerging technologies, these programs become innovative educational proposals that promote the use of technologies [35,36], being a valuable resource that, supported by mobile devices, allows motivation and interest in heritage knowledge [37]. These programs have a diversified educational design typology, such as projects, 3D programs, designs, and educational resources, which allows for an open methodology according to the needs of the context [36]. According to the analysis of quantitative and qualitative studies on heritage education in secondary schools, carried out following the PRISMA method guidelines in systematic reviews, the following objectives can be addressed:

- RQ1: Identify the didactic models used in heritage education teaching.
- RQ2: Examine the areas of educational action and their impact on heritage education teaching.
- RQ3: Analyze the categorization of cultural heritage to determine which learning contents are included in the heritage education of the studies conducted.
- RQ4: Analyze the educational design typologies addressed in heritage education in teaching–learning processes.

## 3. Methodology

A systematic literature review is a scientific investigation that aims to consolidate and objectively integrate a body of research on a specific field of study [38,39]. This research uses a qualitative approach through the execution of methods and techniques that allow for a complete description of the phenomenon under study [40]. To carry out this review, the PRISMA (Preferred Reporting Items for Systematic Reviews and Meta-Analyses) methodology was used, specifically designed for conducting systematic reviews and meta-analyses [41]. The PRISMA methodology consists of a 25-item checklist and a flow diagram that includes four phases: identification, selection, eligibility, and inclusion [42]. The search period for this systematic review spanned from 2018 to 2022, covering a duration of 5 years. The search was conducted from 11 July to 15 July 2022, utilizing the Web of Science, Scopus, and Dimensions databases due to their comprehensive coverage. The present systematic review was developed in three stages: planning, search, and documentation [43].

### 3.1. Planning Stage

In this stage, the topic to be investigated was determined based on the search in Thesauri. When no existing systematic reviews were found, a search protocol was carried out in the Web of Science, Scopus, and Dimensions databases, taking into account the protocol descriptors. Similarly, the criteria used for the selection of databases include high impact and quality in research, coverage, and the breadth of data, as well as thematic

relevance in the educational field, accessibility, and availability. This systematic review is based on the analysis of the following research questions:

- RQ1: What are the didactic models for carrying out heritage education processes?
- RQ2: What are the areas of educational action in heritage education?
- RQ3: What is the categorization of cultural heritage in studies conducted on secondary education students?
- RQ4: What typologies of educational design are addressed in heritage education through teaching–learning processes with secondary students?

From the research questions, the following descriptors and keywords were identified (Table 1) to enhance the precision of study selection.

**Table 1.** Descriptors.

| Descriptor | Synonyms/Keywords in Spanish | Keywords in English |
|---|---|---|
| Descriptor 1: Heritage Education | Educación patrimonial<br>Enseñanza patrimonial<br>Patrimonio cultural | "Heritage education"<br>"Heritage teaching"<br>"Cultural heritage" |
| Descriptor 2: Secondary Education | Enseñanza secundaria<br>Educación secundaria<br>Escuela secundaria | "High school"<br>"Secondary education"<br>"Middle School" |

From the research questions, the following inclusion and exclusion criteria were identified (Table 2) in order to obtain a more precise selection of studies.

**Table 2.** Inclusion and exclusion criteria.

| Inclusion Criteria | Exclusion Criteria |
|---|---|
| - Heritage Education | - Cultural Identity |
| - Secondary Education | - Conference Review |
| - Last 5 Years | - Session Paper |
| - Open-Access Articles | - Book Chapter |

*3.2. Search Stage*

Taking into account the previously stated criteria, we proceeded to formulate the search equations for each of the databases considered in the systematic literature review (Table 3), considering the areas of knowledge.

**Table 3.** Search equations used.

| Databases | Search Equation |
|---|---|
| Web of Science | TS = (("Heritage education" OR "heritage teaching" OR "cultural heritage") AND ("High school" OR "secondary education" OR "Middle School")) |
| Scopus | TITLE-ABS-KEY (("Heritage education" OR "heritage teaching" OR "cultural heritage") AND ("High school" OR "secondary education" OR "Middle School")) |
| Dimensions | (("Heritage education" OR "heritage teaching" OR "cultural heritage") AND ("High school" OR "secondary education" OR "Middle School")) |

A selection of studies was conducted based on a search in the Web of Science, Scopus, and Dimensions databases, using specific search equations. Subsequently, inclusion criteria were applied, which included studies on heritage education in secondary education published in the last 5 years and open-access articles. The information obtained from the databases was organized into a data matrix for systematic analysis.

To carry out this systematic review, the PRISMA method [42] was employed, which is a guide for conducting systematic reviews and meta-analyses. Following this structure, a systematic review of the selected studies was carried out to analyze the literature on heritage education in secondary education. See in Figure 1.

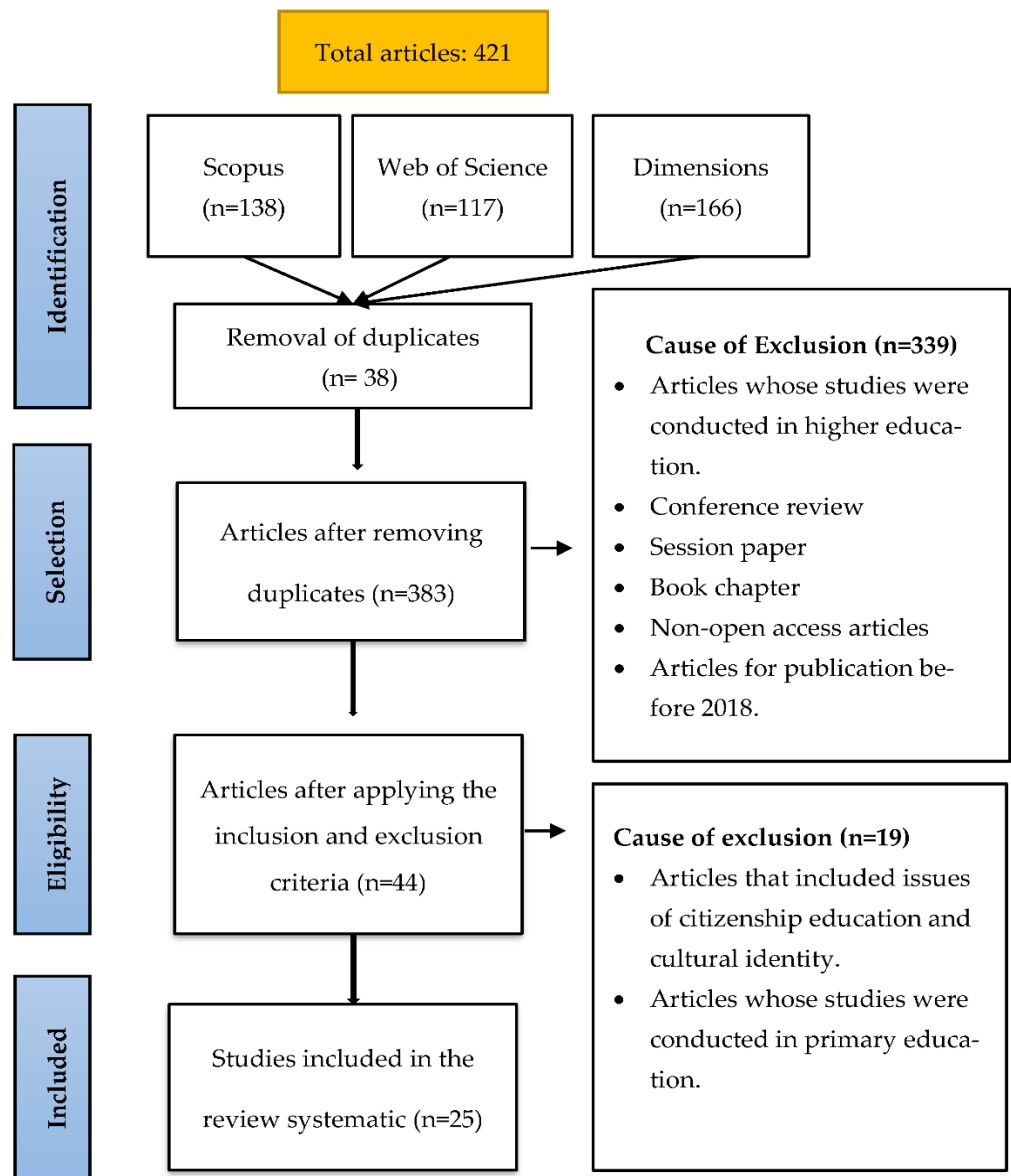

**Figure 1.** Flow of information through the different phases of a systematic review.

## 4. Results

Regarding the year of publication, the selected articles span from 2018 to 2022. From these years, it was observed that 2020 and 2021 had the highest production, with 11 and 5 articles, respectively. This suggests a growing interest in the topic over the last three years. The distribution of the articles over time can be seen in Figure 2.

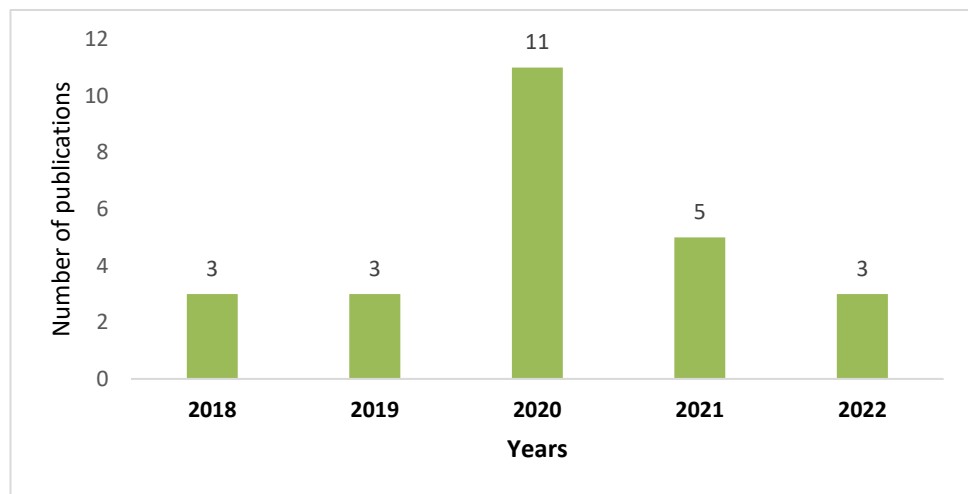

**Figure 2.** Publications per year.

Regarding the country of origin, it is observed that the majority of the selected studies predominantly come from Spain, highlighting a marked geographical concentration in this country, along with significant representation from Indonesia. This phenomenon could be related to various factors, such as language and research networks.

It is important to consider these trends when interpreting the results of the selected studies, which can be seen in Figure 3.

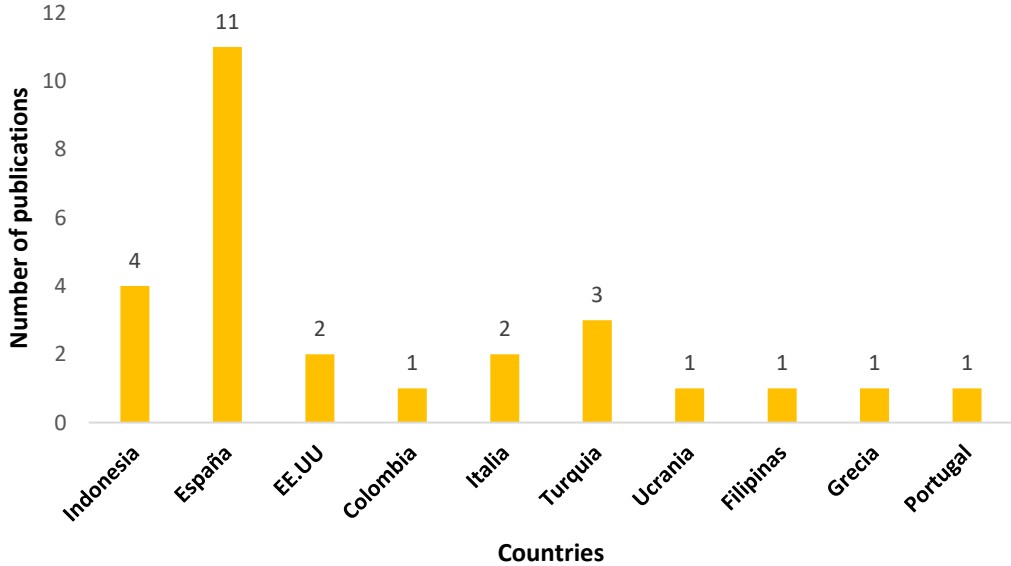

**Figure 3.** Publications per country.

Considering the indicators proposed in the methodology, the results of the research questions according to the analyzed literature are presented as follows:

RQ1: What are the didactic models for carrying out heritage education processes?

In heritage education processes, several relevant characteristics stand out. Regarding context, 21 studies related to classroom interaction and the use of strategies that promote critical and participatory citizenship, as well as the conservation of heritage, were found [30,44]. Regarding content, 9 studies were identified that focus on intangible heritage and show the didactic applications of teachers through oral presentations of legends, stories, and rituals that are part of the cultural identity of students. Teachers use resources such as documentary reviews through written texts, websites, and Android-based applications [45,46]. Regarding teacher behavior, an active, guiding, and innovative role is

highlighted [47,48], coinciding with 18 studies, with the constructivist approach being the most used. In terms of the learner, their protagonism in the learning process is promoted, which was reflected in eight studies that highlight the use of technologies to create digital content, such as videos, images, and the use of geolocation software, among others. These tools not only allow for the acquisition of concepts but also identity and civic values and skills [49,50], as can be seen in Figure 4.

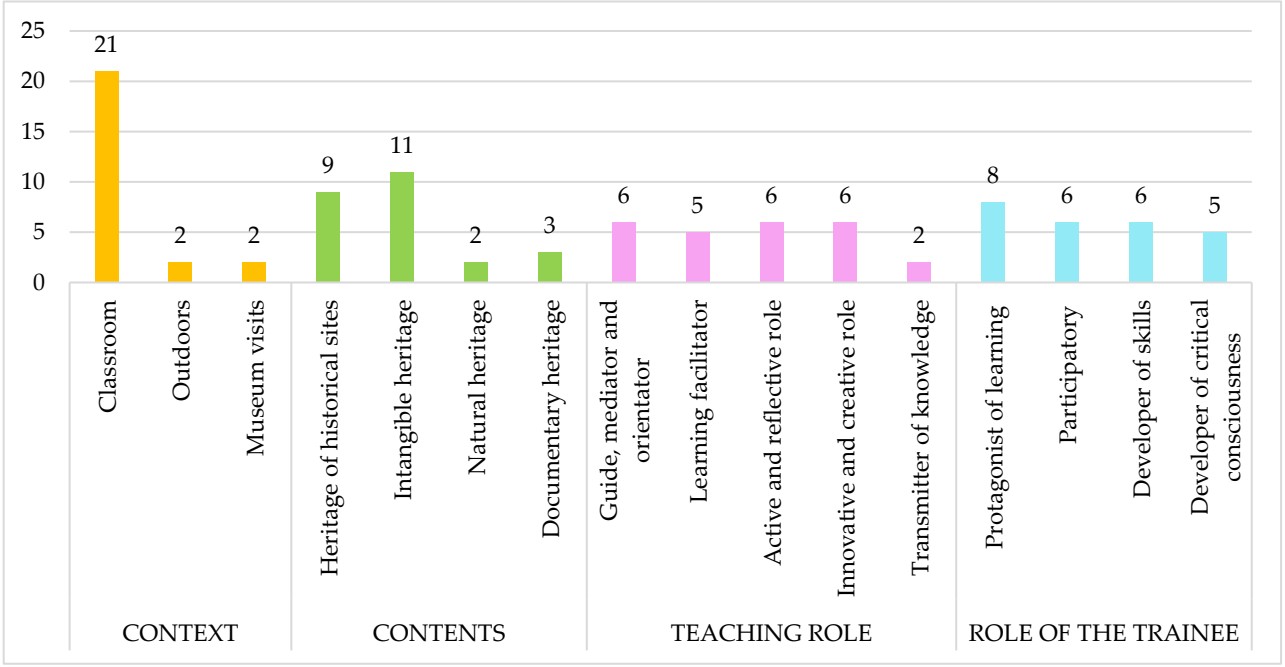

**Figure 4.** Didactic models.

RQ2: What are the areas of educational action in heritage education?

Teachers focus mainly on formal education settings, developing educational projects that involve the entire educational community in order to strengthen identity values and responsible citizenship [51]. To achieve this, they use active methodologies such as problem-based learning and inquiry methods, with the aim of generating a critical attitude and understanding [14,52]. In the informal setting, planning is based on visits to cultural centers such as art galleries, exhibitions, museums, and workshops [48,53], as well as large-scale educational projects such as field trips and fieldwork [50], which promote committed actions towards the conservation and defense of heritage and learning activities both inside and outside the classroom. To a lesser extent, in the nonformal setting, innovative sociocultural animation programs have been proposed, such as Android applications based on virtual tours, educational software programs, and websites [45,54,55], as can be seen in Table 4.

RQ3: What is the categorization of cultural heritage in the studies conducted on secondary education students?

Regarding the heritage categories, studies related to intangible heritage stand out equally. The learning activities in this area are linked to cultural manifestations, as well as visits and excursions that promote interest in their care and preservation [14,30,49–52,54]. On the other hand, some teachers use archaeological sites to guide and develop research projects with their students [3,44,54].

**Table 4.** Areas of educational action.

| Articles | *f* | Area of Action | Description | Forms of Action |
|---|---|---|---|---|
| [2,14,51,52,54,56–60] | 10 | Formal | Formulate projects based on educational management with the objective of taking charge of the different educational levels to promote heritage knowledge. | • Model of educational policies.<br>• Educational projects.<br>• Inquiry method.<br>• Active learning method.<br>• Problem-based method. |
| [3,5,15,30,46,48,50,59] | 8 | Informal | Develop capacities to promote education processes through visits to cultural centers. | • Visits to heritage centers.<br>• Fieldwork.<br>• Inverted classroom.<br>• School trips.<br>• Gamification. |
| [44,45,47,49,54,55,61–63] | 9 | Nonformal | Search for innovations in teaching, the offer of heritage educational programs, and the generation of teaching processes that encourage dialogue between social actors, visitors, and managers. | • Learning based on 360° virtual tours.<br>• Virtual tours.<br>• Software-based learning.<br>• Patrimonializarte program |

Likewise, natural heritage is used pedagogically [57], which develops a spatial vision of territory in students [53]. In the last three years, there has been an increase in technological production related to the use of software, applications, and other digital resources that has allowed the mediation of learning [45,47,50]. However, these resources are used to a lesser extent by teachers, as can be seen in Table 5.

**Table 5.** Heritage categories.

| Articles | *f* | Heritage Categories | Description | Type of Activities |
|---|---|---|---|---|
| [2,14,45,46,49–52,57,58,60] | 11 | Intangible | Cultural expressions, knowledge, and practices that have been passed down from generation to generation. | • Strategies based on the use of websites.<br>• Traditional dance projects. |
| [5,15,30,55,57,61,62] | 7 | Natural heritage | Territorial space focused on environmental aspects. | • Dynamics.<br>• Execution of workshops.<br>• Encourage inquiry processes.<br>• Awareness-raising activities. |
| [2,3,30,44,49,54–56] | 8 | Archaeological sites | Static expression built and occupied in the past | • Application of didactic workshops.<br>• Guided tours. |
| [3,14,30,48,50–52,55,57,58,63] | 11 | Places created by man | Constructions made and designed by contemporary man (theme parks, museums and exhibition centers) | • School field trips.<br>• Visits to museums. |
| [45,47,50,52,54,55,62] | 7 | Digital heritage | Resources that are products of human expression, such as digital texts, images, databases, recordings, web pages or computer programs. | • Website navigation.<br>• 360° virtual tours<br>• Creation of GIS maps.<br>• Creation of virtual museums. |

RQ4: What typologies of educational design are addressed in heritage education from teaching and learning processes in secondary students?

Regarding the educational resources used by teachers, it has been observed that they make use of a variety of materials, such as comics, books, oral sources, films, documentaries, video games, virtual recreations, applications for devices, and websites, with the aim of reconstructing patrimonial knowledge and fostering critical and responsible citizenship in relation to the care and preservation of these places [14,51]. In addition, educational programs have been developed that promote the formation of an environmental conscience and awareness towards the care of heritage [30,56]. Although to a lesser extent, educational projects, educational designs, and workshops are also used that seek significant and innovative activities to arouse the interest of students. It is important to note that, for this purpose, it is necessary that teachers have greater knowledge of the heritage places in their

context and how to manage access to these places involving educational agents, as can be seen in Table 6.

**Table 6.** Typologies of design.

| Articles | *f* | Type of Educational Design | Description | Dynamization |
|---|---|---|---|---|
| [3,50,55,57,59,60] | 6 | Educational project | Learning organization model in which educational agents seek a solution to a perceived problem. | • Museum project.<br>• Schoolwork project.<br>• Project-based learning methodology. |
| [15,30,44,46,48,49,51,56,61] | 9 | Educational program | Set of educational actions to be carried out in order to meet the needs of students and the entire educational community. | • Educational program patrimonializarte.<br>• Educational proposal of educational excursion.<br>• Program of visits to archaeological sites. |
| [2,5,58,63] | 4 | Educational design | Programming of the course, where objectives, activities, and content planning are proposed in order to generate readiness for learning. | • K-12 curriculum.<br>• Heritage awareness curriculum.<br>• Methodology for spatial–visual literacy (MSVL). |
| [14,15,44,45,47,52,54,56,61,62] | 10 | Educational resource | Pedagogical support that reinforces the teacher's performance within the teaching process. | • 360° images.<br>• Videotapes.<br>• Digital libraries and museums.<br>• Android-based application.<br>• Historical GIS maps |
| [15,30,46,59] | 4 | Workshop | Methodology for reflection, experience, and research through collaborative work and learning by discovery. | • Workshop of the senses.<br>• Heritage visit workshop. |

## 5. Discussion

Based on the analysis of the articles, it has been demonstrated that heritage education processes are balanced around four didactic models: context, content, teacher role, and learner role. Regarding the context, learning experiences are mostly carried out in secondary schools, within classrooms, where the teacher plans, executes, and evaluates heritage-focused knowledge. However, the levels of heritage development that should be instilled in the students are not achieved [48,56,57]. In addition, the studies analyzed focus on formative activities within the classroom, unlike other authors who argue that outdoor learning and museum visits promote greater learning by adopting a holistic vision that encompasses history, nature, tradition, and culture, facilitating proximity to the student's environmental and sociocultural reality [5,30,46,48].

In relation to the content-centered teaching model, teachers focus on the development of intangible heritage rooted in their cultural practices, as evidenced in the studies of [2,46]. This promotes a link between students' past and future, developing an awareness of heritage preservation and propagation. Coinciding with the analyzed studies, which focus on intangible cultural heritage associated with their rituals and customs [64], it has been found that the curriculum should be focused on activities centered on archaeological and natural heritage sites, as this contributes to the solution of socionatural problems, with excursions and school visit projects to bring students closer to their heritage, allowing the valorization and care of it [5,30,57]. Likewise, the study by [13] points out that archaeological heritage is of great interest to teacher training, as they consider it an educational tool and a primary source to arouse students' interest.

Both the learner's and teacher's roles are framed in the constructivist approach [51,57]: someone who should be a guide, a mediator, and a learning facilitator [47,49]. However, in the studies by [2,46], traditional strategies based on orality are still used, without method-

ological interventions, which do not allow for the development of a critical attitude toward the preservation of their natural environment and the legacy of the student's ancestors.

Finally, the student-centered didactic model stands out for making the student a key player, a participant, and a builder within their learning process [56,61,62]. Furthermore, the student must develop spatial skills, sensory perception, historical awareness, and leadership abilities to construct their history, achieving an understanding of their heritage. In addition, other studies suggest that they should develop skills to create digital content [45,49,52], where they represent their heritage and convey it within their educational community, fostering identity bonds among students.

In our analysis, we found that teachers frequently plan their learning activities from the formal domain [12,21]. However, it is necessary that the experiences that teachers promote be in both formal and informal contexts, with the support of educational projects and programs that allow students to develop inquiry skills [30,50,54]. In this way, the student is guided to exercise responsible practices and attitudes towards heritage, considering grades, age, and teaching cycle, measured through curricular competencies.

Most of the activities developed fall under the category of intangible heritage, where teachers generate projects, programs, and study plans related to cultural expressions and popular knowledge that are transmitted from generation to generation, deeply rooted in rituals and social uses. These types of activities allow for enhancing a sense of identity and continuity, thus contributing to promoting respect for cultural diversity and human creativity. In this sense, we propose that teachers involve integrative and innovative activities using active methodologies focused on the student, achieving a more comprehensive and critical learning experience [2,56–58]. The generation of new didactic mediation pathways is also suggested through the integration of role-playing [62].

In educational typologies, it has been demonstrated that teachers use educational resources as pedagogical support to reinforce their performance within the teaching process. Materials for teaching are usually considered to be physical; however, there is a growing educational approach that seeks to integrate digital technology for the virtual recreation of a historical monument. This way, a virtual approach to interaction with heritage is encouraged, generating interest in students to know and understand their material and intangible heritage [44,47,54,56].

The first finding shows that didactic proposals focus on intangible heritage inside the classroom [14,52] but fail to integrate material with intangible and natural heritage [2,58,64] to establish a link between historical awareness and preservation that leads to completing the levels of patrimonialization. The second finding indicates that heritage education must encompass formal and nonformal education to achieve a holistic and integrative approach that promotes heritage awareness and citizenship [14] through the development of strategic alliances between the community and the school. The third finding shows that, due to the pandemic, the use of digital educational resources has increased, generating didactic mediation where heritage can be part of a game, an experience, or a knowledge exchange that transports the student to simulated scenarios that enhance learning [64].

Among the limitations of this study, curricular methodological proposals based on simulation and experiences in the role of play in formal, informal, and nonformal education, among others, were not analyzed. The level of involvement that government agencies must fulfill to integrate communities and schools in projects or programs related to heritage from an experiential perspective was not evaluated, as highlighted in most studies [56]. As a recommendation, we suggest analyzing teaching experiences in the design of scenarios, projects, programs, and workshops with a territorial vision to build awareness of the protection and transmission of heritage from a global perspective, with management policies for its implementation in schools to enhance the levels of patrimonialization.

Teachers aspire to initiate teaching–learning processes in the realm of heritage education with their students, yet they seldom make the leap to publication—and it is even more uncommon for such publications to attain a substantial impact factor.

## 6. Conclusions

Ultimately, the purpose of heritage education is to form critical and reflective citizens capable of assuming responsibilities for their territory and promoting sustainability through knowledge, understanding, respect, appreciation, care, enjoyment, and transmission of their historical legacy. In this sense, a systematic review of the literature shows the different didactic models used in heritage education processes, demonstrating that the constructivist approach prevails, in which the teacher and student play complementary roles in activities within the classroom. Despite the use of various teaching strategies and technological resources to achieve a holistic vision of heritage education, it is highlighted that there is still a growing need to incorporate technology in heritage education, leveraging its resources, such as virtual visits, 3D reconstructions, or augmented reality images, to promote motivation and interest in historical and cultural heritage.

However, there is a limitation in heritage education, as educators focus their activities on intangible heritage content based on rituals and customs, strengthening only the identity level and overlooking other expressions. Furthermore, its didactic application follows a traditional approach based on orality.

This demonstrates a difficulty in generating learning itineraries due to a lack of preparation from teachers and limited knowledge in the construction of virtual educational resources and their use. Therefore, it is considered necessary to address teachers' levels of digital training and good educational practices in heritage education regarding the didactic methodologies used to achieve better levels of heritagization.

**Author Contributions:** Conceptualization, Y.K.V.A. and J.L.C.R.; methodology, F.T.-M.; validation, F.H.R.P., Y.K.V.A., J.L.C.R. and F.T.-M.; formal analysis, F.H.R.P. and Y.K.V.A.; research, All; resources, Y.K.V.A. and J.L.C.R.; data curation and F.T.-M.; writing—editing of the original draft, Y.K.V.A., J.L.C.R. and F.T.-M.; drafting—revising and editing, F.H.R.P.; visualization, Y.K.V.A.; supervision, F.H.R.P.; project administration, F.H.R.P. All authors have read and agreed to the published version of the manuscript.

**Funding:** This research received no external funding.

**Data Availability Statement:** Data used and/or analyzed during the present study are available from the corresponding author upon reasonable request.

**Conflicts of Interest:** The authors declare no conflict of interest.

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
