# Peer review of "The Status of Didactic Models for Heritage Education: A Systematic Review"

_heritage, doi:10.3390/heritage6120400_

Round 1

Reviewer 1 Report

Comments and Suggestions for Authors

First of all, congratulations on the research carried out in the field of heritage education in formal contexts, especially with regard to the search for didactic models in secondary education based on the study of scientific production in recent years.

The article, submitted for review, is very well structured and responds perfectly to the skeleton of a research work. All the sections are very well thought out and meet the requirements of presenting an orderly and clear academic work. In this respect, I would like to congratulate you on your writing.

Despite this, I would like to make a series of recommendations that will give the article greater scientific substance and value with a view to its socialisation:

- In the introductory section, it would be interesting to briefly include a definition, as up to date as possible, of the concept of "heritage" and "heritage education". This would allow us to better contextualise the paper's approach.

- Similarly, it would also be interesting to explain what we mean by the term "didactic model", because the meaning differs depending on the country or linguistic area. In this case, as with the previous recommendation, a definition should be included.

- When the period of scientific production that has been analysed under the PRISMA methodology is included, the final year of the analysis (2022) appears, but not the year in which the analysis began (it is assumed that it is 11 June 2017, but it should be written in the article).

- It is curious that in section 4 on the results, it is indicated that the period of analysis is from 2018 to 2022 (being 4 years), and not from 2017 to 2022 (being the five years explained in the methodology: "The search period revolved around the last 5 years, starting the search from July 11th to July 15th, 2022, using the Web of Science, Scopus, and Dimensions databases, since a greater number of studies were found to be analysed" (p. 3). This issue needs to be better resolved [my interpretation is that of the filtered sample of n=25, there are no articles from 2017, which is why it has been limited to 2018-2022; however, it is preferable to make it clearer].

- In the results, when you point out that the countries with the highest production are Spain, Indonesia and Turkey, it is paradoxical that Turkey is mentioned when it has only one more article than Italy or the United States. In this case, I would cite only Spain and Indonesia, and even only the exceptional case of Spain, which far exceeds the rest of the countries.

- In relation to the limitations of the study, it should perhaps be pointed out that a sample of n=25 is certainly too small to make generalisations regarding the current state of heritage education from didactic models. This is partly due to the fact that there is still a lack of publications on heritage education projects, and that on many occasions educational practice goes one way and research goes the other (one of the reasons that should be backed up by authors in order to talk about the lack of studies). Teachers want to activate teaching-learning processes on heritage education with their students, but they rarely take the leap to a publication - and it is even rarer for this publication to have an impact factor.

- The conclusions are repetitive in relation to the results and discussion. It would be interesting to give them a more innovative approach to avoid repetition of ideas. This is a suggestion.

- References: the instructions given by the journal for this section are not followed. A very clear example: there are no italics in the whole section. I invite you to revisit the instructions for authors and adjust your references to what is required by the journal.

- In the references section, I miss some bibliographical reference related to the Faro Convention (2005), a turning point in the treatment of heritage and heritage education from current postulates.

For all these reasons, and to reassess the work carried out, minor changes are recommended for publication. We thank you for thinking of Heritage to publish your work.

Kind regards.

Author Response

Dear Reviewer 1
It is my pleasure to write to you and express my gratitude for the opportunity to review our article, entitled "The Status of Didactic Models for Heritage Education: A Systematic Review". We deeply appreciate your detailed and constructive comments, as well as your acknowledgement of our research in the field of heritage education.
We have thoroughly reviewed each of the suggestions provided, and are pleased to report that we have made the necessary adjustments to improve the quality and accuracy of the article. In the following WORD document, I describe the actions taken in response to each of the comments.

Sincerely yours

Reviewer 2 Report

Comments and Suggestions for Authors

In general, the subject matter of the article is appropriate for the editorial lines of the journal. The citations are correct and up to date. The language is clear and the study can be useful for the scientific community interested in heritage education. 

The text could be improved with the following suggestions:

1.-It is recommended to indicate the criteria for the choice of databases selected for this study. 

-The exclusion criteria that mention "Primary education" or "Articles whose studies were conducted in higher education" could be eliminated, because  there is an inclusion criterion, which is the secondary education stage. The same applies to the exclusion criterion for non-open articles.

-Table 3 is not necessary. The information shown does not need to be expressed through this resource. It can be indicated by a paragraph in the text. 

-In the description of the sample or in the conclusions, the biases of the sample and the biases of this research should be explained. 

-In the discussion, statements are made about teachers' teaching practices that are not sufficiently scientifically supported. A study with a larger sample or focus on this aspect would be needed to substantiate claims about teachers' teaching practices. 

-In the references there are some works that refer to infant and primary education that could be removed. 

Author Response

Dear Reviewer 2
It is my pleasure to write to you and express my gratitude for the opportunity to review our article, entitled "The Status of Didactic Models for Heritage Education: A Systematic Review". We deeply appreciate your detailed and constructive comments, as well as your acknowledgement of our research in the field of heritage education.
We have thoroughly reviewed each of the suggestions provided, and are pleased to report that we have made the necessary adjustments to improve the quality and accuracy of the article. In the following WORD document, I describe the actions taken in response to each of the comments.

Sincerely yours

Reviewer 3 Report

Comments and Suggestions for Authors

There is a mistake in line 62: "...formulated by the [a word is missing]".

In line 257, I wonder if would be better "Heritage places" instead "patrimonial places".

Author Response

Dear Reviewer 3
It is my pleasure to write to you and express my gratitude for the opportunity to review our article, entitled "The Status of Didactic Models for Heritage Education: A Systematic Review". We deeply appreciate your detailed and constructive comments, as well as your acknowledgement of our research in the field of heritage education.
We have thoroughly reviewed each of the suggestions provided, and are pleased to report that we have made the necessary adjustments to improve the quality and accuracy of the article. In the following WORD document, I describe the actions taken in response to each of the comments.

Sincerely yours
